# One-Step Calcination to Gain Exfoliated g-C_3_N_4_/MoO_2_ Composites for High-Performance Photocatalytic Hydrogen Evolution

**DOI:** 10.3390/molecules27217178

**Published:** 2022-10-24

**Authors:** Yan Chen, Ao Li, Xiuli Fu, Zhijian Peng

**Affiliations:** 1School of Science, China University of Geosciences, Beijing 100083, China; 2School of Science, Beijing University of Posts and Telecommunications, Beijing 100876, China

**Keywords:** g-C_3_N_4_, MoO_2_, exfoliation, photocatalytic hydrogen evolution

## Abstract

The difficulty of exposing active sites and easy recombination of photogenerated carriers have always been two critical problems restricting the photocatalytic activity of g-C_3_N_4_. Herein, a simple (NH_4_)_2_MoO_4_-induced one-step calcination method was successfully introduced to transform bulk g-C_3_N_4_ into g-C_3_N_4_/MoO_2_ composites with a large specific surface area. During the calcination, with the assistance of NH_3_ and water vapor produced by ammonium molybdate, the pyrolytical oxidation and depolymerization of a g-C_3_N_4_ interlayer were accelerated, finally realizing the exfoliation of the g-C_3_N_4_. Furthermore, another pyrolytical product of ammonium molybdate was transformed into MoO_2_ under an NH_3_ atmosphere, which was in situ loaded on the surface of a g-C_3_N_4_ nanosheet. Additionally, the results of photocatalytic hydrogen evolution under visible light show that the optimal g-C_3_N_4_/MoO_2_ composite has a high specific surface area and much improved performance, which is 4.1 times that of pure bulk g-C_3_N_4_. Such performance improvement can be attributed to the full exposure of active sites and the formation of abundant heterojunctions. However, with an increasing feed amount of ammonium molybdate, the oxidation degree of g-C_3_N_4_ was enhanced, which would widen the band gap of g-C_3_N_4_, leading to a weaker response ability to visible light. The present strategy will provide a new idea for the simple realization of exfoliation and constructing a heterojunction for g-C_3_N_4_ simultaneously.

## 1. Introduction

With the development of human society, energy crises and environmental problems have caused more and more troubles for mankind, and semiconductor photocatalysis technology is a promising technology to solve these two problems [1,2]. For better photocatalytic effects, a variety of semiconductor materials have been developed as photocatalysts. Among them, g-C_3_N_4_ has attracted extensive attention because of its suitable band gap structure, stable physical and chemical properties, environmental friendliness, low cost, and easy availability [3]. In addition, because g-C_3_N_4_ has a graphene-like structure (that is, a two-dimensional layered structure), it theoretically has a large specific surface area, which can lead to a huge number of reactively active sites in theory and act as a framework carrier [4]. However, during its thermal polymerization, due to the disordered growth and ubiquitous intermolecular interaction between the layers, the synthesized g-C_3_N_4_ is usually highly concentrated as lumps, called bulk g-C_3_N_4_ by scholars, which can only expose a small specific surface area, preventing its photocatalytic reaction [5]. Therefore, how to peel g-C_3_N_4_ from layer to layer to expose more active sites is a hot topic in this field.

In the literature, a lot of researchers have tried to prepare exfoliated g-C_3_N_4_. At the outset, the Hummers’ method for the preparation of graphene oxide was used to exfoliate g-C_3_N_4_ [6]. Thereupon, lots of chemical delamination methods using a large number of strong acids and strong oxidants were developed [7,8,9,10,11]. However, because such methods should apply a large number of strong acids and/or oxidants such as concentrated H_2_SO_4_, HNO_3_, HCl, H_3_PO_4_, and KMnO_4_, which impose great harm on the environment, greener methods have to be developed. Almost at the same time, methods using ultrasonic separation with low surface-tension solvents have been developed. For example, Lin et al. acquired g-C_3_N_4_ nanosheets by ultrasonic and centrifugation accompanied with a variety of organic solvents [12]. Although g-C_3_N_4_ nanosheets can be obtained more environmentally friendly by such a method, it cannot be widely used in practical application due to its high equipment requirements and very low productivity. In addition, Niu et al. found that secondary calcination could effectively exfoliate bulk g-C_3_N_4_, because the principle of its interlayer separation is the oxidation reaction and depolymerization of g-C_3_N_4_ under an atmosphere of high temperature [13]. Moreover, Wu et al. adopted a method by adding a large amount of NH_4_Cl in combination with secondary calcination to obtain exfoliated g-C_3_N_4_ nanosheets at a lower temperature [14].

However, only gaining exfoliated g-C_3_N_4_ cannot solve the problems that limit the enhancement of its photocatalytical performance. As is well known, the rapid recombination of photogenerated carriers is another critical factor that seriously affects the photocatalytic efficiency of g-C_3_N_4_. To solve this problem, excepting the deposition of materials with high electrical conductivity onto the surface of g-C_3_N_4_, loading semiconductor materials onto the surface of g-C_3_N_4_ nanosheets to form heterojunctions is a classical program, because it is also conducive to optoelectronic transmission on the two-dimensional layer-like structured g-C_3_N_4_ [15,16,17].

Among semiconductor materials, MoO_2_ has excellent electrical conductivity and photoelectrochemical activity due to its unique metallic properties which are attributed to the fermi level of MoO_2_ provided by its 4d orbital [18,19]. Therefore, MoO_2_ has been widely investigated and applied in photocatalysis, electrocatalysis, batteries, and other fields [20,21,22]. Recently, researchers have reported some work on g-C_3_N_4_/MoO_2_ composites for photocatalysis. In the literature, Li et al. realized an obvious improvement in photocatalytic efficiency in the reduction of CO_2_ over the g-C_3_N_4_/MoO_2_ composite, which can be attributed to the construction of a Schottky junction between g-C_3_N_4_ and MoO_2_ in the composite [23]. Yadav et al. prepared a microspherical MoO_2_/g-C_3_N_4_ core-shell composite for the high-efficiency photocatalytic degradation of xylenol orange by a simple one-step calcination procedure [24]. In addition, Ji et al. prepared a g-C_3_N_4_/MoO_2_ 2D-2D structured composite by a cumbersome three-step process for enhanced photocatalytic degradation of Rhodamine B [25]. However, there are very few studies on the photocatalytic hydrogen evolution of g-C_3_N_4_/MoO_2_ composites [18]. Consequently, it is necessary to further explore the possibility of g-C_3_N_4_/MoO_2_ composites in photocatalytic hydrogen evolution.

In this work, a novel g-C_3_N_4_/MoO_2_ composite with a large specific surface area was successfully prepared by employing a simple one-step calcination method for high-performance photocatalytic hydrogen evolution. In the whole process of this method, NH_3_ and water vapor produced by the pyrolysis of ammonium molybdate break the van der Waals forces between the layers of g-C_3_N_4_, which then lets O_2_ in the air flow to promote the oxidation and depolymerization process of g-C_3_N_4_ between the layers and grow MoO_2_ particles onto the surface of the g-C_3_N_4_ interlayer, ultimately realizing the spalling of g-C_3_N_4_ and the construction of heterojunctions between g-C_3_N_4_ and MoO_2_ through a single calcination. The gained g-C_3_N_4_/MoO_2_ composite has much improved photocatalytic performance for hydrogen evolution under visible light, which can be attributed to the effectively enhanced specific surface area by the exfoliation of bulk g-C_3_N_4_ and the reduced recombination of photogenerated carriers by forming the heterojunctions of g-C_3_N_4_ with MoO_2_. The photocatalytic hydrogen evolution rate (HER) over the optimal sample is 4.1 times greater than that over bulk g-C_3_N_4_. In addition, the photocatalytic mechanism of the obtained g-C_3_N_4_/MoO_2_ composite was proposed. The current strategy will provide a new idea for simple exfoliation while also constructing a heterojunction for g-C_3_N_4_.

## 2. Results and Discussion

### 2.1. Composition, Structure, and Morphology

In order to examine the morphological features of the samples, the samples were observed by SEM. As can be seen in Figure 1a, g-C_3_N_4_ directly obtained by calcination from melamine shows a large bulk structure, which is due to the stacking of g-C_3_N_4_ layers caused by the interlayer van der Waals force. After thermal treatment in air, the g-C_3_N_4_ sample presents a measure of spalling owing to the corrosion on the edge of g-C_3_N_4_ by oxygen at high temperatures. From Figure 1b, it can be clearly seen that the g-C_3_N_4_ bulk became smaller after thermal treatment without any other change in morphology. However, the sample treated with ammonium molybdate solution, as shown in Figure 1c,d, displays a more obvious lamellar structure, which can be attributed to the escape of NH_3_ from the thermal decomposition of ammonium molybdate. In addition, it can be seen from Appendix A in the Appendix A that with increasing amounts of the applied ammonium molybdate, the size of the resultant g-C_3_N_4_ nanosheets gradually decreases.

In order to further examine the morphology and elemental composition of the samples, TEM characterization was carried out. Compared with the sample CNH (see Figure 1e), the sample treated with ammonium molybdate exhibits a clearer exfoliation between the layers (Figure 1f). Meanwhile, TEM mapping scanning was performed on the area of Figure 1f, and the results are shown in Figure 1g. As is seen, four elements of C, N, O, and Mo were identified by the system. Among them, the distributions of C and N elements are consistent with the shape of g-C_3_N_4_ nanosheets as shown in Figure 1f. However, the Mo and O elements are only of a small amount and evenly distributed on the surface of the g-C_3_N_4_ nanosheets. In order to find out the scattering feature of molybdenum oxide on the g-C_3_N_4_ nanosheets, with a small piece of sample at this position, high-resolution TEM (HRTEM) characterization was carried out. Through the HRTEM image (Figure 1h), it can be observed that some quantum dots with a diameter of about 2–3 nm are distributed on the surface of the nanosheets. Although their crystallinity is not very fine, a small number of lattice stripes can still be identified on the quantum dots, which could be indexed to the MoO_2_ (110) crystal plane. From the above results, it can be inferred that the thermally treated g-C_3_N_4_ is effectively exfoliated into thinner nanosheets, and a small amount of MoO_2_ quantum dots are loaded on the surface of the g-C_3_N_4_ nanosheets.

Typical XRD patterns of the bulk g-C_3_N_4_, CNH, and CNM sample are shown in Figure 2a. It can be seen from this figure that all the samples contain the two characteristic peaks of g-C_3_N_4_. The peak around 13.2° is indexed to the (100) crystal plane of g-C_3_N_4_ (0.671 nm), which represents the tri-s-triazine repeating unit in the g-C_3_N_4_ plane, while the peak around 27.6° is attributed to the (002) crystal plane of g-C_3_N_4_ (0.323 nm), signifying the interlayer stacking conjugated aromatic system of g-C_3_N_4_ in the vertical direction [13,16]. Remarkably, compared with those of bulk g-C_3_N_4_, the strengths of these two peaks of the other samples are relatively low. This result can be interpreted as the spalling between g-C_3_N_4_ layers and plane fracturing, which would cause the structural damage of the g-C_3_N_4_ crystal [12]. Obviously, due to the interlayer separation caused by the escape of NH_3_, the four composite samples from CNM-1 to CNM-8 have weaker XRD peaks, and this fact further verifies that the processes presented in this work can peel g-C_3_N_4_ effectively. However, the peak of MoO_2_ cannot be identified from these patterns because MoO_2_ in these samples exists in the form of quantum dots with a high degree of dispersion, poor crystallinity, and a small molar fraction. In order to confirm the phase composition of molybdenum oxide in the samples, a large number of composite samples were collected from the Al_2_O_3_ ceramic boat after the synthesis processes, and another 3 h of thermal treatment under air was performed, which was intended to thermally decompose g-C_3_N_4_ for enriching molybdenum oxide. The enriched samples were characterized by XRD, and the result is shown in Figure 2b. It can be clearly seen in this figure that the peaks corresponding to the MoO_2_ phase are consistent with standard PDF card no. 78-1072.

In order to investigate the elemental composition and chemical state of the composite samples, XPS analysis on them was performed. Since the XPS spectra of the prepared series of composite samples are similar, only the spectra of the optimal sample CNM-4 are displayed. Through the XPS survey spectrum (see Appendix A), it can be seen that the composite sample is composed of C, N, Mo, and O. From the high-resolution spectra (Figure 3), the chemical state of each element in the composite can be further understood. In the C 1s spectrum (Figure 3a), two peaks centered at the binding energies of 284.77 and 288.22 eV can be observed in the sample, which are attributed to the free carbon for instrument calibration and the C element in N-C=N, respectively [26]. For comparison, the high-resolution C 1s spectrum of bulk g-C_3_N_4_ was also presented in this figure. It is seen that this spectrum has the same peaks at similar positions as that of the optimal sample CNM-4. The only difference between these two spectra is that the binding energy of CNM-4 at 288.22 eV is 0.18 eV higher than that of bulk g-C_3_N_4_, due to the inductive effect of the O atoms, which are connected to the g-C_3_N_4_ structure caused by the thermal oxidation process. In addition, in the N 1s spectrum of the sample CNM-4 (Figure 3b), four peaks with binding energies of 398.69, 399.34, 401.05, and 404.35 eV can be fitted, representing N atoms in C-N=C, N-(C)_3_, and C-N-H structures in the g-C_3_N_4_ framework as well as free N_2_ adsorbed from air, respectively [26]. Similarly, the peaks in the N 1s spectrum representing the g-C_3_N_4_ structure in the sample CNM-4 are shifted to the high value of binding energies by about 0.3 eV compared with those of bulk g-C_3_N_4_, which is due to the inductive effect of the O atoms connected with the g-C_3_N_4_ structure. On the basis of the chemical states of C and N elements discussed above, it can be concluded that the composite does contain g-C_3_N_4_. As for the Mo 3d spectrum of the optimal sample CNM-4 (Figure 3c), there are three groups of peaks with binding energy at 230.30 and 233.31 eV, 231.36 and 235.30 eV, as well as 232.27 and 236.37 eV, representing the electron spin splitting of 3d_5/2_ and 3d_3/2_ of Mo^4+^, Mo^5+^, and Mo^6+^ in the composites, respectively [20,27]. The XPS spectrum of the Mo element revealed that the oxidation state of Mo atoms in the composites involved Mo^4+^, Mo^5+^, and Mo^6+^, with the Mo^5+^ and Mo^6+^ states attributed to the Jahn–Teller effect caused by lattice distortion [28]. This result is consistent with the XPS Mo spectrum of pure MoO_2_, also shown in Figure 3c. Thus, it can be concluded that the molybdenum oxide in the optimal sample CNM-4 is composed of MoO_2_. From the XPS spectrum of O 1s (Figure 3d), three peaks centered at the binding energies of 520.21, 531.98, and 533.31 eV could be observed, which can be assigned to the oxygen element in MoO_2_, the N-C-O structure, and free H_2_O, respectively [27]. Compared with the XPS spectrum of bulk g-C_3_N_4_, the O peak area representing N-C-O in the sample CNM-4 displays an obvious increase, which can be attributed to the O atoms introduced into the g-C_3_N_4_ structure by thermal oxidation.

In addition, the content of each element in the prepared samples in this work was calculated based on XPS analysis and is listed in Table 1. It can be seen that C and N elements account for the main components in each sample, and the atomic ratio of C to N is close to 3:4. Thus, it is believable that the main component of these samples is g-C_3_N_4_. Furthermore, the content of Mo in these samples increased gradually as the concentration of ammonium molybdate in the precursor solution used for the thermal treatment increased. Additionally, the content of O element is clearly increased, which is more than twice the content of Mo element, indicating that, in addition to the O atoms in the introduced MoO_2_, there are additional O atoms in the samples. This result should be attributed to the O atoms introduced into g-C_3_N_4_ that is combined directly with the g-C_3_N_4_ structure in the samples by the oxidation reaction during the thermal treatment in air. In addition, the increase in oxidation degree reflects the destruction of the intermolecular force between g-C_3_N_4_ layers by the gases produced by the decomposition of ammonium molybdate. In other words, when O_2_ can enter the g-C_3_N_4_ layers smoothly, the oxidation process will be further accelerated. In summary, through the XPS analysis of the composite samples, it can be concluded that such samples are composed of O-doped g-C_3_N_4_ and MoO_2_.

To further explore the internal pore structure in the composite samples, a set of N_2_ adsorption-desorption tests were conducted on bulk g-C_3_N_4_, CNH, and this series of CNM composite samples. The recorded N_2_ adsorption-desorption isotherms are shown in Figure 4a. Based on them, the specific surface area (S_BET_), pore diameter, and pore volume of the samples were calculated. The results are listed in Table 2, in which the pore diameter distribution is also displayed in Figure 4b. Firstly, comparing the S_BET_ values of the samples (see Table 2) with that of the bulk g-C_3_N_4_ (12.34 m^2^·g^−1^), the sample CNH shows a larger S_BET_ value of 55.65 m^2^·g^−1^, which can be attributed to the oxidation of g-C_3_N_4_ by O_2_ from air [13]. The S_BET_ values of CNM-1, CNM-2, CNM-4, and CNM-8 samples were 102.91, 89.67, 96.17, and 116.32 m^2^·g^−1^, respectively, which were much larger than those of CN and CNH samples. Based on the above facts, the increase in specific surface area can be attributed to the spalling effect of ammonia produced by the decomposition of ammonium molybdate in the g-C_3_N_4_ layers and O_2_ corrosion of g-C_3_N_4_ [14]. Then, the pore size distribution of the samples was examined (see Figure 4b). Compared with that of the bulk g-C_3_N_4_, the CNH sample has more pores with a large pore size of about 10~50 nm but fewer pores with a small pore size of roughly 3.833 nm, indicating that partial small pores are expanded into large mesopores under the erosion of O_2_. The change of small pores into large pores will lead to an increase in specific surface area and pore volume. Moreover, the pore diameter of the sample CNM-1 increases in both ranges, which can be explained by the fact that the escape of NH_3_ produces a large number of micro/mesopores and oxygen corrosion simultaneously. Furthermore, with an increasing concentration of ammonium molybdate in the precursor solution, the pore diameter distribution changed significantly. After examining the pore diameter of CNM-2, CNM-4, and CNM-8, it can be seen that more small-pore-size mesopores and fewer large-pore-size micropores are built in the samples. This phenomenon may be attributed to two factors. On the one hand, because the escape of more ammonia reduced the local oxygen concentration in the sample, the oxidation of g-C_3_N_4_ was limited, thereby restraining the further expansion of pores. On the other hand, after more MoO_2_ was deposed into the large pore, part of the large pore was blocked into small ones. As a result, there are more small pores and fewer large pores after adding more (NH_4_)_2_MoO_4_·4H_2_O.

### 2.2. Synthesis Strategy and Formation Mechanism

According to the above characterization results of the samples, the synthesis strategy and formation mechanism of the CNM composite samples are proposed in Figure 5. As shown in Figure 5a, initially, bulk g-C_3_N_4_ was dispersed in ammonium molybdate solution under stirring in order to fully adsorb MoO_4_^2−^ and NH_4_^+^ on the g-C_3_N_4_ sheet. Considering that the radius of NH_4_^+^ is 0.142 nm and that of MoO_4_^2−^ is 0.192 nm, as well as from XRD data, the calculated radius of the small ring in the g-C_3_N_4_ sheet is 0.134 nm and the diameter of the large pot is 0.671 nm, the NH_4_^+^ and MoO_4_^2−^ ions in the precursor solution can only enter into the large pot in the g-C_3_N_4_ sheet [29,30], as shown in Figure 5b. Then, after centrifugation, the precipitate was collected and dried, obtaining the precursor for the preparation of the target sample. Finally, the precursor was placed in an Al_2_O_3_ ceramic boat and heated under an air atmosphere in a tube furnace to obtain the target sample. According to thermogravimetric analysis data of the precursors (see Appendix A), the samples with ammonium molybdate began to lose weight at around 450 °C and completely decomposed after 600 °C, whereas the original g-C_3_N_4_ sample began to lose weight after 500 °C and completely decomposed at 700 °C. Therefore, it can be deduced that the presence of ammonia would reduce the decomposition temperature and increase the decomposition rate of g-C_3_N_4_. Based on these facts, the formation mechanism of the present CNM samples can be proposed as follows (Figure 5b). MoO_4_^2−^ and NH_4_^+^ ions entered into the interlayer of bulk g-C_3_N_4_ and further entered into the large pot in the g-C_3_N_4_ lamellas. In the process of thermal treatment, the ammonium molybdate absorbed between the g-C_3_N_4_ nanosheets is decomposed. During this process, NH_4_^+^ is transformed into ammonia, and the escape of ammonia will destroy the van der Waals forces between g-C_3_N_4_ molecular layers, finally leading to exfoliated layers of g-C_3_N_4_. On the other hand, the amount of ammonia generated in the g-C_3_N_4_ nanosheets cannot diffuse out immediately and forms a reductive environment. Thus, MoO_4_^2−^ was decomposed into MoO_3_ initially, which was then converted into MoO_2_ at 500 °C in the reductive ammonia atmosphere [28]. At the same time, O_2_ in the air flow diffuses into the g-C_3_N_4_ interlayer and oxidizes g-C_3_N_4_, forming numerous pores of larger size in g-C_3_N_4_ (see Figure 5c). Finally, the proposed g-C_3_N_4_/MoO_2_ composites are formed.

### 2.3. Photocatalytic H_2_ Evolution Performance

The photocatalytic performance was evaluated by measuring the photocatalytic hydrogen evolution ability over the samples under visible light, and the results are displayed in Figure 6. Figure 6a clearly shows that the HER of the samples CNH and CNM-1 to CNM-8 increases with increasing ammonium molybdate concentration in the pre-cursor solution, reaching a maximum value of 320.8 μmol·g^−1^·h^−1^ when the ammonium molybdate concentration is 4 mmol·L^−1^. However, under the same conditions, the photocatalytic HER of a bulk g-C_3_N_4_ sample is only 78.4 μmol·g^−1^·h^−1^. In other words, the performance of the optimal sample CNM-4 in this work is 4.1 times that of bulk g-C_3_N_4_. Therefore, it can be confirmed that the improved performance of the composite samples is due to the increasing amount of MoO_2_ in the samples and their enhanced specific surface area. Compared with CNM-4, the photocatalytic HER of the sample CNM-8 would be significantly reduced because the band gap of C_3_N_4_ in CNM-8 reaches 3.02 eV, which can only absorb light with a wavelength below 410 nm. However, because a 400 nm filter was used to cut off the UV light source below 400 nm in this work, the shortage of the available light for the sample CNM-8 would certainly result in the decline of its hydrogen evolution performance. Consequently, although a higher concentration of ammonium molybdate can be beneficial for both enhancing the amount of MoO_2_ and producing more pores in the sample CNM-8, the intensified oxidation of bulk g-C_3_N_4_ will widen the band gap of g-C_3_N_4_, making it impossible to make effective use of visible light. In addition, compared with bulk g-C_3_N_4_, although the sample CNH will lose partial absorption capacity of visible light sources due to the oxidation of g-C_3_N_4_, its photocatalytic hydrogen evolution performance was still improved significantly. This result can be attributed to the exposure of more active sites caused by the expansion of a specific surface area. The as-prepared composite samples thermally treated with ammonium molybdate show more excellent photocatalytic hydrogen evolution performance because of their larger specific surface area and more abundant mesopores in the samples. Based on the above results, it can be concluded that the improvement of photocatalytic hydrogen evolution performance of the present g-C_3_N_4_/MoO_2_ composite samples is caused by a synergistic effect of the MoO_2_ recombination with g-C_3_N_4_ and the increase in specific surface area. However, it has to be noted that excessive oxidation will reduce the light absorption range of g-C_3_N_4_, which will ultimately finally lead to a decline in the photocatalytic hydrogen evolution performance of the samples.

Moreover, whether a photocatalyst can work stably for a long time is also very important for its practical application. Therefore, cyclic stability tests were performed for the optimal sample CNM-4 under the same conditions as described above. Notably, the cyclic stability tests consist of 5 rounds of a 3 h photocatalytic hydrogen evolution process, for which the system should be re-evacuated after each run. From the test results shown in Figure 6b, it can be concluded that the sample CNM-4 has good stability for photocatalytic hydrogen evolution, since the sample can still maintain stable, high hydrogen evolution efficiency after five rounds of tests.

### 2.4. Photoelectrochemical Properties

In order to characterize the absorption ability of the prepared materials for light, the UV-Vis absorption spectra of bulk g-C_3_N_4_, CNH, and all the g-C_3_N_4_/MoO_2_ composite samples were measured. Figure 7a shows the recorded spectra of these samples. As shown in the figure, when compared to bulk g-C_3_N_4_, after thermal treatment, the resultant samples CNH and CNM would have a lower light absorption capacity due to their widened band gap. Moreover, Figure 7b displays their corresponding (ahν)^2^ versus E_g_ plots, from which the band gap of the corresponding materials can be obtained. As can be seen, the band gap of the sample bulk g-C_3_N_4_, CNH, and CNM-1–CNM-8 is 2.78, 2.88, 2.86, 2.94, 2.96, and 3.02 eV, respectively, which can be attributed to the changed structure caused by the increased degree of oxidation on g-C_3_N_4_. The band gap of g-C_3_N_4_ is excited by π-π * excitation, in which the conjugated C 2p and N 2p orbitals in the heptazine form the conduction band, while the 2p orbitals of the edge N atoms form the valence band [31]. However, the oxidation of g-C_3_N_4_ will break the C=N bond conjugated structure into an N-C-O bond, resulting in an expanded band gap of g-C_3_N_4_ in the sample. Although the band gap of the samples CNM-1–CNM-4 continues to increase, their light absorption capacity becomes stronger and stronger, which can be attributed to the increased loading amount of MoO_2_ in the sample. On the other hand, the sample CNM-8 shows a poor light absorption capacity due to the greatly widened band gap of g-C_3_N_4_ caused by excessive oxidation.

To explain the photocatalytic activity for hydrogen evolution of the prepared samples, their photoelectric response ability and carrier transport ability were investigated by transient photocurrent response and electrochemical impedance tests. Figure 8a shows the transient photocurrent response curves of all the samples. From this figure, it can be clearly seen that the optimal sample CNM-4, which presents the highest efficiency for photocatalytic hydrogen evolution, also has the strongest photocurrent response ability. It is easy to understand that its photocurrent intensity is stronger than those of the samples CNM-1 and CNM-2 because it contains more MoO_2_. Because the loaded MoO_2_ can effectively separate photogenerated electrons and holes, more free electrons will be converted into current without becoming internal energy after the compositing of MoO_2_ with g-C_3_N_4_. However, the photocurrent intensity of CNM-8 with the highest content of MoO_2_ is attenuated, which is obviously due to the poor light utilization ability of CNM-8, because it can only make use of the light in the band range between 400–410 nm after the light with a wavelength shorter than 400 nm is filtered. In addition, it is seen that the photocurrent intensity of bulk g-C_3_N_4_ is stronger than that of the sample CNH, because the widening of the band gap after the oxidation of g-C_3_N_4_ will result in a poor utilization efficiency of light.

The EIS Nyquist plots of all the samples obtained from the electrochemical impedance test are shown in Figure 8b. Obviously, the impedance of all the samples after thermal treatment decreased with the increase of the MoO_2_ content in them. This phenomenon shows again that the compositing of MoO_2_ with g-C_3_N_4_ is conducive to the transfer of photogenerated carriers through the photocatalyst. Additionally, this result can be attributed to the following two reasons: firstly, MoO_2_ has better electrical conductivity than g-C_3_N_4_, and another reason is that the compositing of g-C_3_N_4_ with MoO_2_ forms a built-in electric field, which is also conducive to charge transmission. The resultant lower impedance is beneficial to improving the transmission efficiency of photogenerated carriers, preventing them from recombination, effectively improving the photocatalytic performance. In addition, the impedances of CNH and CNM-1 are higher than that of bulk g-C_3_N_4_, which is because the impedance depends on both g-C_3_N_4_ and MoO_2_. The impedance of g-C_3_N_4_ would be increased after it was oxidized, while the deposition of MoO_2_ with good electrical conductivity would decrease the impedance of the samples. The impedance of the samples is actually a balanced result between the oxidation of g-C_3_N_4_ and the deposition of MoO_2_. Consequently, the impedance of CNH is higher than that of bulk g-C_3_N_4_ due to the oxidation of g-C_3_N_4_. The impedance of CNM-1 is higher than that of bulk g-C_3_N_4_ because the deposition of a small amount of MoO_2_ cannot compensate for the counteracting effect of the oxidation of g-C_3_N_4_. In summary, the increase in MoO_2_ content in the composites can effectively reduce the impedance of the samples, while the oxidation of g-C_3_N_4_ will increase the impedance of the samples. Therefore, it is the superposition of the two factors that determines the final impedance of the samples.

### 2.5. Photocatalytic Mechanism

Based on the results mentioned above and the support of relevant theories, the photocatalytic hydrogen evolution mechanism over the obtained photocatalysts was proposed in Figure 9. In order to obtain the energy band correlation between g-C_3_N_4_ and MoO_2_ in the figure, with the optimal sample CNM-4 as an example, Mott-Schottky plots, VB-XPS spectra, and UV-Vis absorption spectra were carried out, and the results are displayed in Appendix A. In this study, the flat band potentials of CNM-4 and MoO_2_ can be obtained from the recorded Mott-Schottky plots, which are −0.53 and −0.52 eV, respectively. After eliminating the influence of the potential from the Ag/AgCl reference electrode, their final flat band potential values are −0.31 and −0.30 eV, respectively [32]. From the VB-XPS spectra, the potential differences in the energy bands of CNM-4 and MoO_2_ from the valence band top to the Fermi level are 2.02 and 0.23 eV, respectively [33]. In general, the flat band potential value is approximately equal to the Fermi level [32,34]. Based on the above data, it can be calculated that the valence band values of CNM-4 and MoO_2_ are 1.72 and −0.07 eV, respectively. In addition, the band gap of CNM-4 and MoO_2_ could be obtained from Figure 7b and Appendix A, and the calculated values of their valence bands are −1.24 and −1.80 eV, respectively. Finally, according to the energy band data obtained from the test, the photocatalytic hydrogen evolution mechanism was proposed.

As is seen, three factors are considered to influence the photocatalytic hydrogen evolution performance of g-C_3_N_4_/MoO_2_ composite. The first is the heterojunction between g-C_3_N_4_ and MoO_2_, which can form a built-in electric field there and promote the separation of photogenerated electrons and holes, preventing them from recombination. Under the irradiation of visible light, the photocatalyst was excited to generate lots of free electrons and holes. The photogenerated electrons in the conduction band of g-C_3_N_4_ pass across the heterojunctions through the connected Fermi level due to the energy difference between the two sides of the heterojunction into MoO_2_, which can combine with the holes in the valence band of MoO_2_, where a depletion layer is formed. As a result, the counteraction of electrons was prevented, thus retaining the holes in the valence band of g-C_3_N_4_ and the electrons in the conduction band of MoO_2_. These free photogenerated electrons will further rapidly move to the Pt particles deposited on the surface of the catalysts and attend to the redox reaction with H^+^ in the solution to produce hydrogen. Meanwhile, the photogenerated holes in g-C_3_N_4_ can react with TEOA to be quenched. Of course, the transfer process of carriers from conduction band to conduction band and/or from valence band to valence band in such a Type II heterojunction may also be in question. Nevertheless, if the electrons in the valence band of g-C_3_N_4_ are transferred to the valence band of MoO_2_, the total oxidation potential of the catalyst will decrease dramatically. As a result, it is unable to react with TEOA. So, such a process was almost impossible during the photocatalytic process over the present composite catalyst. In other words, the formation of heterojunctions promotes the photocatalytic hydrogen evolution of the present composite catalyst.

The second factor is the exfoliation of g-C_3_N_4_, which can effectively increase the specific surface area and form numerous mesopores in the catalyst. A larger specific surface area is conducive to the exposure of active sites in a catalyst, and the formation of more mesopores is advantageous to the adsorption of reaction substrates and the release of the formed hydrogen. This point explains the improvement in the photocatalytic hydrogen evolution performance of the samples CNM-1–CNM-8.

Finally, the oxidation of g-C_3_N_4_ during thermal treatment causes the π-π conjugated structures in the heptazine group to be destroyed, which will widen the band gap. With the increasing oxidation degree, the band gap of the samples increases gradually. As a result, the band wavelength of visible light that can be used for photocatalytic hydrogen evolution will be narrowed, ultimately degrading the photocatalytic performance of the sample.

In summary, the photocatalytic hydrogen evolution performance of the present g-C_3_N_4_/MoO_2_ samples is affected by three factors: the loading amount of MoO_2_, the extent of exfoliation, and the degree of g-C_3_N_4_ oxidation. Among them, high MoO_2_ content and full exfoliation are conducive to photocatalysis, but the oxidation of g-C_3_N_4_ is also intensified with increasing content of MoO_2_ in the sample. Therefore, the sample CNM-4 instead of CNM-8 is the optimal sample for photocatalytic hydrogen evolution by adjusting the synthesis and application conditions.

## 3. Experimental Details

### 3.1. Raw Materials

Ammonium molybdate ((NH_4_)_2_MoO_4_, 99.0%) and sodium sulfate anhydrous (Na_2_SO_4_, 99.0%) were bought from Sinopharm Chemical Reagent Co., Ltd. (Shanghai, China). Melamine (99.0%), triethanolamine (TEOA, 98.0%), and molybdenum dioxide (MoO_2_, 99.0%) were bought from Aladdin Reagent Co., Ltd. (Shanghai, China). Potassium hexachloroplatinate (K_2_PtCl_6_, 99.9%) was purchased from Macklin Biochemical Co., Ltd. (Shanghai, China). All the raw materials are used without further treatment in this work.

### 3.2. Preparation of g-C_3_N_4_

The g-C_3_N_4_ sample was synthesized by calcining melamine in the air [26]. Typically, 10 g of melamine was placed into an Al_2_O_3_ ceramic crucible with a half cover and the calcination was carried out in a muffle furnace at 550 °C for 4 h. After the calcination, a bulk g-C_3_N_4_ sample could be obtained, and by grinding the collected bulk sample in a mortar, g-C_3_N_4_ powder was ready for use.

### 3.3. Preparation of CNM Nanocomposites

The composite samples of g-C_3_N_4_ and MoO_2_ (CNM) were synthesized by a one-step calcination method. Typically, a certain amount of ammonium molybdate powder was dissolved in 100 mL of deionized water. Then, 1 g of bulk g-C_3_N_4_ powder was added into the prepared ammonium molybdate solution and dispersed for 1 h under stirring. Afterwards, a precursor could be obtained from the sediment by centrifuging and following drying. Finally, the precursor was placed in an Al_2_O_3_ ceramic boat and heated under an air atmosphere at 500 °C for 2 h in a tube furnace to obtain the target sample. The samples obtained by thermally treating g-C_3_N_4_ powder with different concentrations of ammonium molybdate solution (1, 2, 4, and 8 mmol/L) were named CNM-1, CNM-2, CNM-4, and CNM-8, respectively. In addition, a sample using the same process for the preparation of CNM samples but without ammonium molybdate in deionized water was also prepared as the control sample (CNH).

### 3.4. Materials Characterization

The morphology and microstructure of the samples were observed by a field-emission scanning electron microscope (FE-SEM, Gemini SEM 500, Carl Zeiss, Oberkohen, Germany) and transmission electron microscope (TEM, FEI Tecnai G2 F20 U-TWIN, Hillsborough, OR, USA). The elemental compositions in the samples were examined by the energy dispersive X-ray spectroscope attached to the TEM. The phase composition of the samples was identified by X-ray diffraction (XRD, D/max-RB, Tokyo, Japan; Cu Kα radiation, λ = 1.5418 Å) in a continuous scanning mode at a speed of 8°·min^−1^. An X-ray photoelectron spectroscope (XPS, EscaLab MKII, Thermo Fisher Scientific, Waltham, MA, USA; C 1s of 284.8 eV as reference) was utilized to investigate the elemental composition and chemical state. An Agilent Varian Cary 5000 UV-Vis spectrometer (Palo Alto, CA, USA) was used to record the UV-Vis absorption spectra of the samples by using solid powders. N_2_ adsorption-desorption curves were measured by using a Quantachrome Autosorb-IQ adsorption instrument (Boynton Beach, FL, USA). In addition, the specific surface area and pore-size distribution of the samples were evaluated by the multipoint Brunner–Emmet–Teller (BET) and Barret–Joyner–Halenda (BJH) methods, respectively. A Linseis STA PT1600 thermogravimetric analyzer (TGA, Zelbu, Germany) was used to collect the data for thermogravimetric analysis.

### 3.5. Evaluation of Photocatalytic H_2_ Evolution

The hydrogen evolution test of each sample was carried out on a Labsolar IIIAG automatic photocatalytic system from PerfectLight (Beijing, China). During the test, 50 mg of sample was first dispersed in 100 mL of 10 vol.% triethanolamine (TEOA) aqueous solution. Then, a small amount of K_2_PtCl_6_ equivalent to 1 wt.% of the tested sample was added into the system for the deposition of Pt onto the sample surface by Xe lamp illumination for 30 min. Afterwards, the photocatalytic hydrogen evolution performance of the samples was evaluated under the illumination of an Xe lamp for 3 h. All the light sources were provided with a 300 W Xe lamp equipped with a 400 nm UV cut-off filter and 20 cm away from the top of the reactor. The testing temperature was maintained at 7 °C by circulating cooling water to prevent boiling. The resultant gas was analyzed by gas chromatography (Techcomp GC-7900, Shanghai, China), in which the carrier gas was N_2_. The cyclic tests were repeated five times under the same test conditions.

### 3.6. Electrochemical Measurements

The working electrode employed in the electrochemical tests was prepared by 50 μL of ink coating on a FTO glass substrate, which was composed of the as-prepared powder photocatalyst (5 mg), 20 μL of 5 wt.% Nafion ethanol solution, and 500 μL of ethanol. The area covered by the ink on the FTO glass was 1 cm^2^. Additionally, the electrolyte solution was a 0.2 M Na_2_SO_4_ aqueous solution.

All the related electrochemical tests were performed on a CHI 660E electrochemical workstation by using a three-electrode system, in which the counter electrode was Pt foil, and the reference electrode was Ag/AgCl. The light source was provided by a 300 W Xe lamp equipped with an UV cut-off filter (λ ≥ 400 nm).

## 4. Conclusions

An exfoliated g-C_3_N_4_/MoO_2_ composite was synthesized by a simple and effective (NH_4_)_2_MoO_4_-induced one-step calcination method. Specifically, through the thermal decomposition of (NH_4_)_2_MoO_4_, the gaseous product NH_3_ and water vapor between the g-C_3_N_4_ layers can expand the layer spacing and thus the oxygen in the air flow can accelerate the oxidation and depolymerization process of g-C_3_N_4_, finally realizing the exfoliation of g-C_3_N_4_, while the solid product MoO_3_ is transformed into MoO_2_ in the NH_3_ atmosphere, finally loading on the g-C_3_N_4_ surface. Such exfoliated g-C_3_N_4_/MoO_2_ composites show an effectively improved photocatalytic hydrogen evolution performance, in which the optimal g-C_3_N_4_/MoO_2_ composite has a high specific surface area and presents a high HER (320.8 μmol·g^−1^·h^−1^) under visible light, reaching 4.1 times that of bulk g-C_3_N_4_. The improvement in photocatalytic activity of the present g-C_3_N_4_/MoO_2_ composites can be attributed to the full exposure of active sites caused by the enhanced specific surface area and the formation of numerous mesopores in the catalysts, as well as the formation of abundant heterojunctions between g-C_3_N_4_ and MoO_2_, which alleviates the recombination of photogenerated carriers. In addition, it should be noted that the oxidation of g-C_3_N_4_ will widen the band gap of g-C_3_N_4_ and reduce its utilization ability in visible light. The present strategy will provide a new idea for the simple realization of exfoliation and constructing a heterojunction for g-C_3_N_4_ simultaneously.

## Figures and Tables

**Figure 1 molecules-27-07178-f001:**
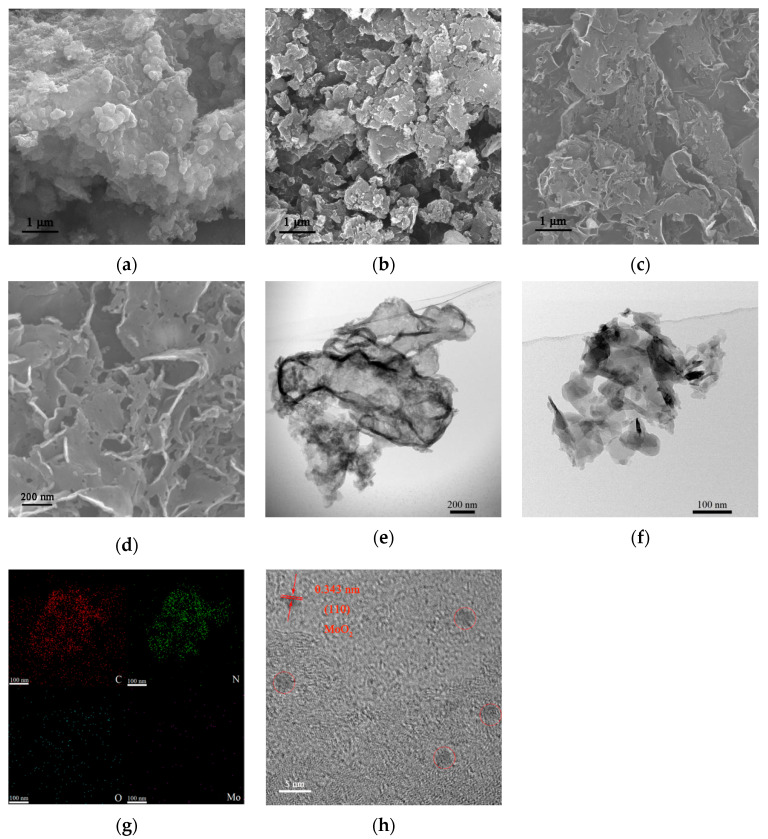
Microstructures of the prepared samples. Typical SEM images: low-magnification for (**a**) bulk g-C_3_N_4_ and (**b**) CNH, and (**c**,**d**) different, higher magnification for CNM-4. Typical TEM images of (**e**) CNH and (**f**) CNM-4. EDX elemental mapping (**g**) and HRTEM (**h**) images corresponding to CNM-4.

**Figure 2 molecules-27-07178-f002:**
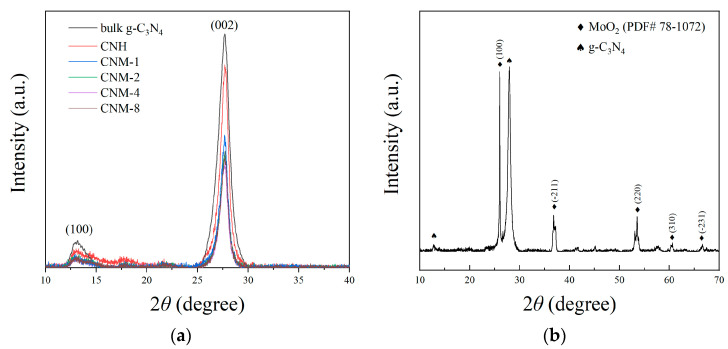
XRD results: (**a**) comparison on the patterns of CNM, CNH, and bulk g-C_3_N_4_, and (**b**) single pattern for the optimal sample CNM-4 thermally treated under air for another 3 h.

**Figure 3 molecules-27-07178-f003:**
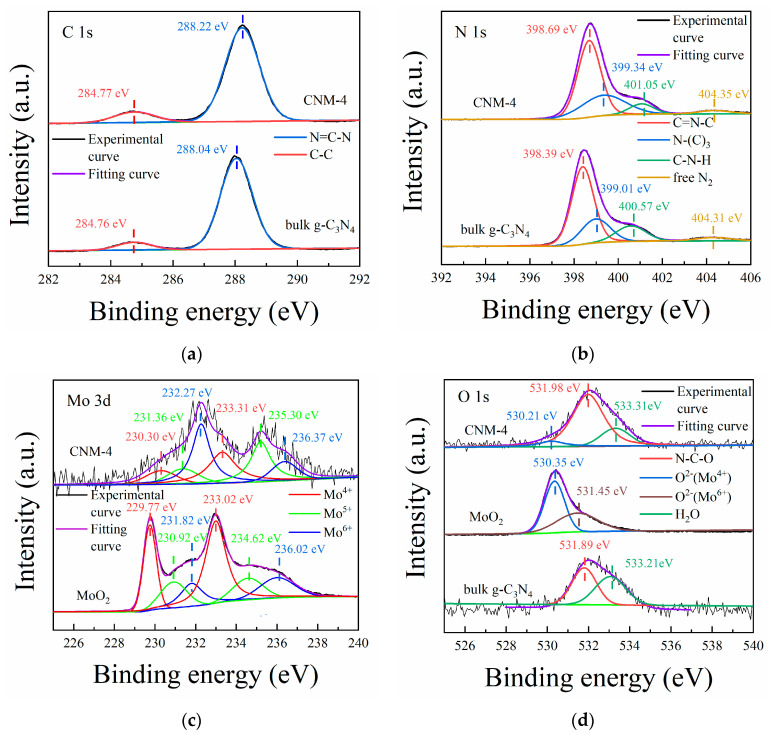
XPS spectra of the optimal composite CNM-4 in comparison with those of the control samples (bulk g-C_3_N_4_ and commercial MoO_2_): (**a**) C 1s, (**b**) N 1s, (**c**) Mo 3d, and (**d**) O 1s.

**Figure 4 molecules-27-07178-f004:**
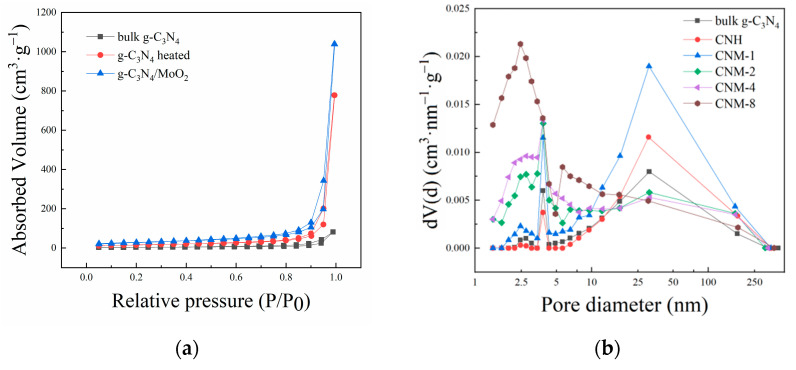
(**a**) N_2_ adsorption-desorption isotherm and (**b**) pore diameter distribution of the prepared samples.

**Figure 5 molecules-27-07178-f005:**
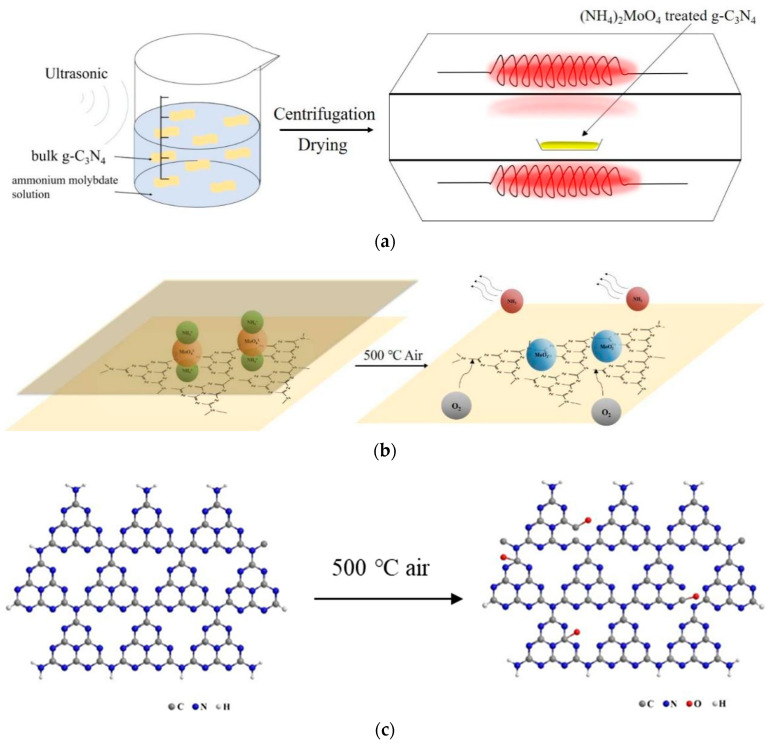
Schematic diagrams of (**a**) the synthesis strategy for the samples, (**b**) the formation mechanism for the exfoliated g-C_3_N_4_/MoO_2_ composite, and (**c**) the oxidation mechanism for g-C_3_N_4_.

**Figure 6 molecules-27-07178-f006:**
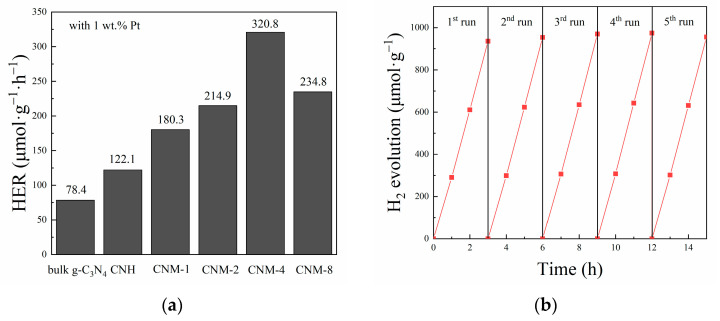
(**a**) Photocatalytic H_2_ evolution rate of bulk g-C_3_N_4_, CNH, and all exfoliated g-C_3_N_4_/MoO_2_ composites; (**b**) cycling performance for H_2_ evolution over the optimal CNM-4 composite.

**Figure 7 molecules-27-07178-f007:**
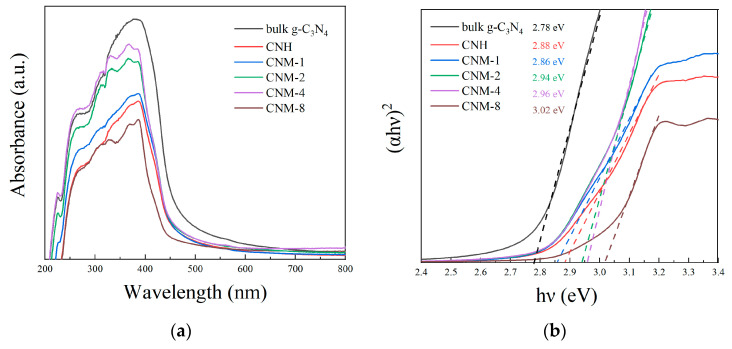
(**a**) UV-Vis absorption spectra of bulk g-C_3_N_4_, CNH, and all g-C_3_N_4_/MoO_2_ composites, and (**b**) their corresponding (ahν)^2^ versus E_g_ plots.

**Figure 8 molecules-27-07178-f008:**
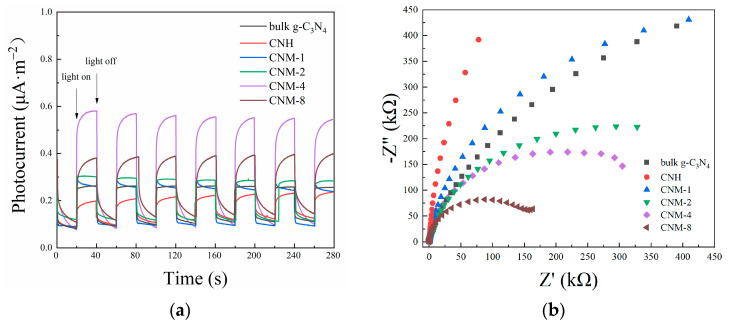
Photoelectrochemical results of bulk g-C_3_N_4_, CNH, and all the exfoliated g-C_3_N_4_/MoO_2_ composites: (**a**) transient photocurrent response curves and (**b**) EIS Nyquist plots.

**Figure 9 molecules-27-07178-f009:**
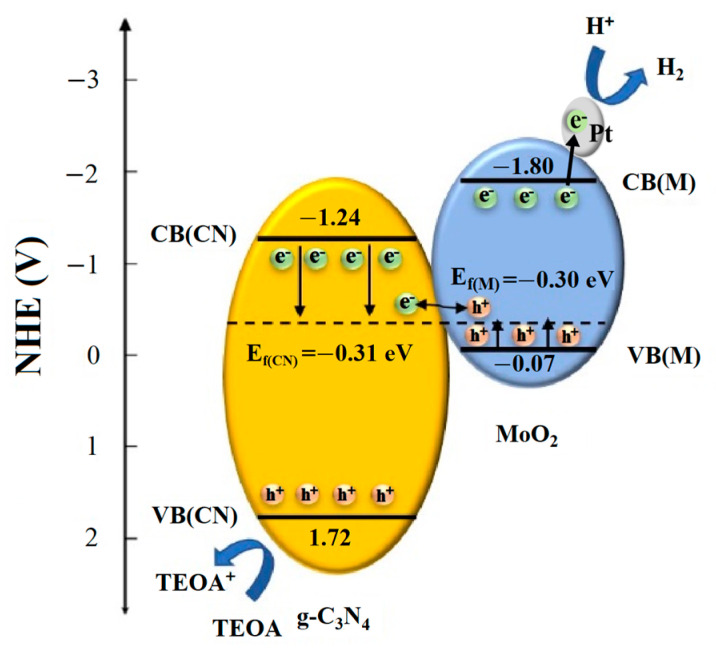
Schematic diagram of the photocatalytic HER mechanism over the g-C_3_N_4_/MoO_2_ composites.

**Table 1 molecules-27-07178-t001:** Elemental compositions of each sample.

Samples	C (at.%)	N (at.%)	Mo (at.%)	O (at.%)
bulk g-C_3_N_4_	42.76	56.57	/	0.67
CNH	42.92	55.69	/	1.39
CNM-1	42.80	55.11	0.09	2.00
CNM-2	42.56	54.42	0.15	2.87
CNM-4	42.32	54.21	0.26	3.21
CNM-8	42.07	53.32	0.43	4.18

**Table 2 molecules-27-07178-t002:** S_BET_, pore volume, and pore diameter of the prepared samples.

Samples	S_BET_ (m^2^·g^−1^)	Pore Volume (cm^3^·g^−1^)	Average Pore Diameter (nm)
bulk g-C_3_N_4_	12.34	0.129	3.833
CNH	55.65	1.213	30.816
CNM-1	102.91	1.618	30.072
CNM-2	89.67	1.136	3.833
CNM-4	96.17	1.225	3.833
CNM-8	116.32	0.848	2.455

## Data Availability

Not applicable.

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
