# Peer review of "One-Step Calcination to Gain Exfoliated g-C3N4/MoO2 Composites for High-Performance Photocatalytic Hydrogen Evolution"

_molecules, 2022, doi:10.3390/molecules27217178_

Round 1
Reviewer 1 Report
In this work, the authors reported a one-step calcination method for preparation of g-C3N4/MoO2 composite, which displayed enhanced catalytic performance than parent materials. The structure was studied properly in detail, and the observed formation mechanism was interesting. The system in principle can be further expanded to other composites. Therefore, I recommend its acceptance after the following minor points:
1. Scale bars in Figure 1e,f are not clear, and missing in Figure 1g.
2. Reference should be included in line 207-209.
3. The y-axis in Figure 4a should be "Absorbed volume"; The y-axis in Figure 6b should be "evolution"; Figure S2 x-axis is "binding"?
4. In line 309-312, is that possible that less large pore in CNM-8 is the reason of more MoO2 deposition and pore blockage?
5. The space after "g-C3N4" in line 374, 403 and 511 is subscript?
6. In line 393, "resultant"; 454, "AgCl"; 471, "a free and holes";
7. The active site for H2 generation is Pt? Would the Pt deposition help to explain the mechanism? The composites still maintain high performance without Pt?
Reviewer 2 Report
This manuscript by Prof. Peng et al. describes the preparation of g-C3N4/MoO2 hybrid materials and the investigation of HER (hydrogen evolution reaction) performance of the corresponding materials. Bulk g-C3N4 is reported to be effectively exfoliated by the introduction of (NH4)2MoO4.4H2O, which gives NH3 for the subsequent exfoliation and chemical reactions of g-C3N4. The hybrid materials are characterized. The HER performance for the materials is examined and the possible photocatalytic HER mechanism is discussed.
There are some critical issues to be addressed prior to the acceptance by the Journal.
(1) The author claims that the formation of g-C3N4/MoO2 hybrid materials from the thermal treatment of g-C3N4 and (NH4)2MoO4.4H2O. It is known that (NH4)2MoO4.4H2O is thermally decomposed to o-MoO3 under air. In this study, the Mo end product in air is MoO2 instead of MoO3. Additional explanation is needed.
(2) This photocatalytic HER is conducted under visible light. It is recommended to clearly indicated.
(3) In the Experimental section, 1% wt of K2PtCl6 is added to the materials for the HER study. Apparently, the actual catalytic site is Pt not MoO2. The schematic presentation in Fig 9 is incorrect.
(4) After the HER, is there any Mo leaching?
(5) The material composition from the enrichment by thermal decomposition may not be the same with that in the pre-enriched samples.
(6) The XPS results of CNM-4 reveal a set of peaks at 231.90 and 235.43 eV. For the 3d5/2 of Mo6+, the energy of 231.90 eV is a little bit low. The author may need to re-simulate the data or give a different assignment.
(7) The author has used amorphous MoO2 for the XPS characterization (Fig 3c). In fact, it is known that amorphous MoO2 contains a variety of Mo-oxide species bearing Mon+ (n = 4, 5, 6). It is strongly recommended to re-check the XPS using pure MoO2 for the comparison.
(8) In Fig 4a, which CNM-x is g-C3N4/MoO2?
(9) The elemental composition for all samples containing Mo can show deviations in different batches. The standard deviation should be shown to display the consistence of the material composition.
(10) The explanations to the changes of pore sizes and volumes are not clear. The paragraph needs to be re-phrased for clarity.
(11) The exfoliation mechanism is based on the presence of NH3 while the NH3 from (NH4)2MoO4.4H2O is generated on the surface of the g-C3N4 sheets. Additional explanation is needed.
(12) In Fig S3, the TGA data of CHN is to be added.
(13) For UV-vis results (Fig 7a), what is the concentration for each sample? Is there any suspension in the solution?
(14) The transient photocurrent response results of all samples are inconsistent with the photocatalytic HER performance. For example, CHN has the lowest photocurrent and the highest resistance while the HER performance of CNH is greater than that of bulk g-C3N4. Additional explanations and experiments are needed.
Round 2
Reviewer 2 Report
The author has made all possible changes in this revision. This Reviewer is satisfied by the changes. The revision is recommended for publication.